# Amphetamine reduces reward encoding and stabilizes neural dynamics in rat anterior cingulate cortex

**Saeedeh Hashemnia, David R Euston, Aaron J Gruber***

Canadian Center for Behavioral Neuroscience, Department of Neuroscience, University of Lethbridge, Lethbridge, Canada

**Abstract** Psychostimulants such as d-amphetamine (AMPH) often have behavioral effects that appear paradoxical within the framework of optimal choice theory. AMPH typically increases task engagement and the effort animals exert for reward, despite decreasing reward valuation. We investigated neural correlates of this phenomenon in the anterior cingulate cortex (ACC), a brain structure implicated in signaling cost-benefit utility. AMPH decreased signaling of reward, but not effort, in the ACC of freely-moving rats. Ensembles of simultaneously recorded neurons generated task-specific trajectories of neural activity encoding past, present, and future events. Low-dose AMPH contracted these trajectories and reduced their variance, whereas high-dose AMPH expanded both. We propose that under low-dose AMPH, increased network stability balances moderately increased excitability, which promotes accelerated unfolding of a neural 'script' for task execution, despite reduced reward valuation. Noise from excessive excitability at high doses overcomes stability enhancement to drive frequent deviation from the script, impairing task execution.

*For correspondence:
aaron.gruber@uleth.ca

**Competing interests:** The authors declare that no competing interests exist.

## Introduction

Animals can draw from a large repertoire of innate and learned actions to generate behavioral output (*Whishaw and Kolb, 2005*). Even well-trained rodents sometimes engage in sleep, grooming, exploration, or other behaviors during laboratory tasks. When hungry, however, food rewards usually have sufficient value to motivate task engagement in lieu of these other options. Effort, reward, and other factors pertinent for decisions about resource collection are often formalized within the concept of utility (*Phillips et al., 2007*; *Glimcher et al., 2009*). Options requiring low effort and yielding a large food reward have high utility to hungry animals, whereas options requiring high effort or yielding little/unwanted food have low utility. Animals often seek to maximize utility, even if it requires exerting additional effort. For instance, rodents and primates typically choose to exert increased effort if the associated reward is of considerably higher value than that of lower-effort options (*Salamone et al., 1994*; *Hosokawa et al., 2013*). This preference is plastic. Moderate doses of d-amphetamine (AMPH) increases the amount of work rodents will exert for reward (*Floresco et al., 2008b*; *Bardgett et al., 2009*). It also increases engagement in learned food-seeking behaviors (*Foltin, 2001*; *Odum and Shahan, 2004*). The most obvious explanation for these effects according to utility theory is that AMPH increases the perceived value of reward. Behavioral data, however, suggest that reward valuation is *decreased*. AMPH-treated rats are less motivated to eat; their latency to first consumption is longer, they consume less than usual, and they spend less time eating (*Blundell et al., 1976*; *Blundell et al., 1979*; *Leibowitz et al., 1986*). Suppression of food-intake by AMPH is also evident in primates (*Foltin, 2001*). Why do animals work harder for food they appear less motivated to consume? It is possible that AMPH somehow affects the perceived cost of effort, or that increased task engagement is a by-product of motoric hyperactivity.

But neither of these provide a robust explanation for the full suite of effects, including the drastic shift of responding to grooming and stereotyped outputs at high doses (*Randrup et al., 1963*; *Randrup and Munkvad, 1967*). The behavioral data, therefore, do not support a clear prediction about how AMPH may influence the neural encoding of utility.

Cost-benefit decisions engage a network of brain structures, and the anterior cingulate cortex (ACC) appears particularly important for those involving physical effort (*Rudebeck et al., 2006*; *Floresco et al., 2008a*). The preference of rats for high-effort, high-reward options is reduced or eliminated by ACC lesions (*Walton et al., 2002*; *Walton et al., 2003*; *Schweimer and Hauber, 2005*; *Holec et al., 2014*). Electrophysiological recordings indicate that rat ACC and nearby regions in the medial prefrontal cortex (mPFC) encode a variety of signals related to choice and task execution. These include the position of the animal (*Euston and McNaughton, 2006*; *Fujisawa et al., 2008*; *Mashhoori et al., 2018*), task phase (*Lapish et al., 2008*; *Balaguer-Ballester et al., 2011*), reward (*Gruber et al., 2010*; *Cowen et al., 2012*), choice (*Cowen et al., 2012*), effort (*Cowen et al., 2012*; *Hashemniayetorshizi et al., 2015*), and other features (*Cowen and McNaughton, 2007*; *Gruber et al., 2009*; *Durstewitz et al., 2010*; *Sul et al., 2010*). Some ACC neurons also jointly encode costs-benefit information, which is well suited to utility signaling (*Hillman and Bilkey, 2010*; *Cowen et al., 2012*). These data are consistent with findings in monkeys and humans (*Isomura et al., 2003*; *Kennerley et al., 2006*; *Croxson et al., 2009*; *Skvortsova et al., 2014*; *Blanchard et al., 2015*; *Klein-Flügge et al., 2016*). Although AMPH clearly modulates ACC activity (*Lapish et al., 2015*), its effect on the encoding of effort-reward utility and other task variables in ACC has not been explicitly shown. Here, we attempt to link these independent observations to better understand how AMPH affects ACC encoding and dynamics pertinent to task engagement and outcome valuation.

## Results

We used high-density electrophysiology to record ensembles of single neuron activity from the ACC of well-trained rats performing a continuous version of the classic T-maze (*Figure 1A*). Rats ran from a starting feeder to one of two target feeders accessible after turning to the right or left at the choice point. Rats were forced to alternate to the left and right sides on subsequent trials, and trials were organized into blocks in which either the reward volume delivered at the target feeders, or the effort (barrier climb) required to reach each target feeder, was different (*Figure 1B*). Rats performed five blocks of trials, then received an injection of either saline or AMPH, and performed the five blocks again. We quantified changes in behavior and neural signaling after the injection, with respect

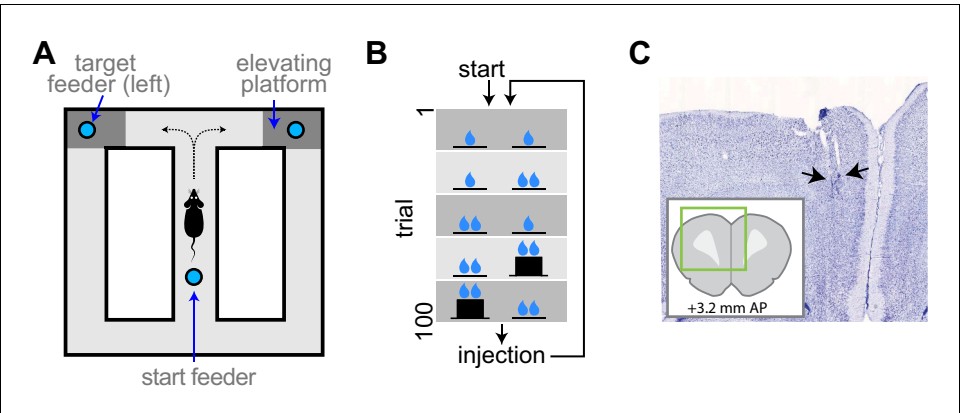

**Figure 1.** Experimental apparatus and task. (**A**) Illustration of the figure 8 maze. The target feeders are elevated on some trials to impart physical effort. (**B**) Schematic illustration of the task. The reward volume is indicated by the number of drop symbols, and the climbing effort by the black trace of feeder platform height. The sequence of trial blocks (background shading) is repeated after drug injection. (**C**) Representative sample of a histological section showing the endpoints of electrodes (arrows). The inset show the relative position of the section aligned to a standard rat brain atlas (*Paxinos and Watson, 2014*).

to the pre-injection phase. We recorded a total of 1209 putative pyramidal neurons from 22 session (55 ± 6.7 simultaneous cells per session) in four rats.

## AMPH increases running speed and decreases reward consumption time

AMPH administration in our animals evoked typical locomotor effects in a dose-dependent manner. It increased the median running speed (*Figure 2A*; Kruskal-Wallis, $\chi^2(3)>109$; p<0.0001). This effect peaked at 1.0 mg/kg because rats became more frequently disengaged in the task at higher doses. Indeed, the amount of off-task behavior (circling, pausing, backtracking) increased with dose (*Figure 2B*; main effect ANOVA, $F_{3,18}$ = 3.36; p=0.042; power = 0.76). Off-task behaviors became so prominent at 2.0 mg/kg that animals did not complete a sufficient number of trials for analysis. Data at this concentration is therefore excluded from the present report.

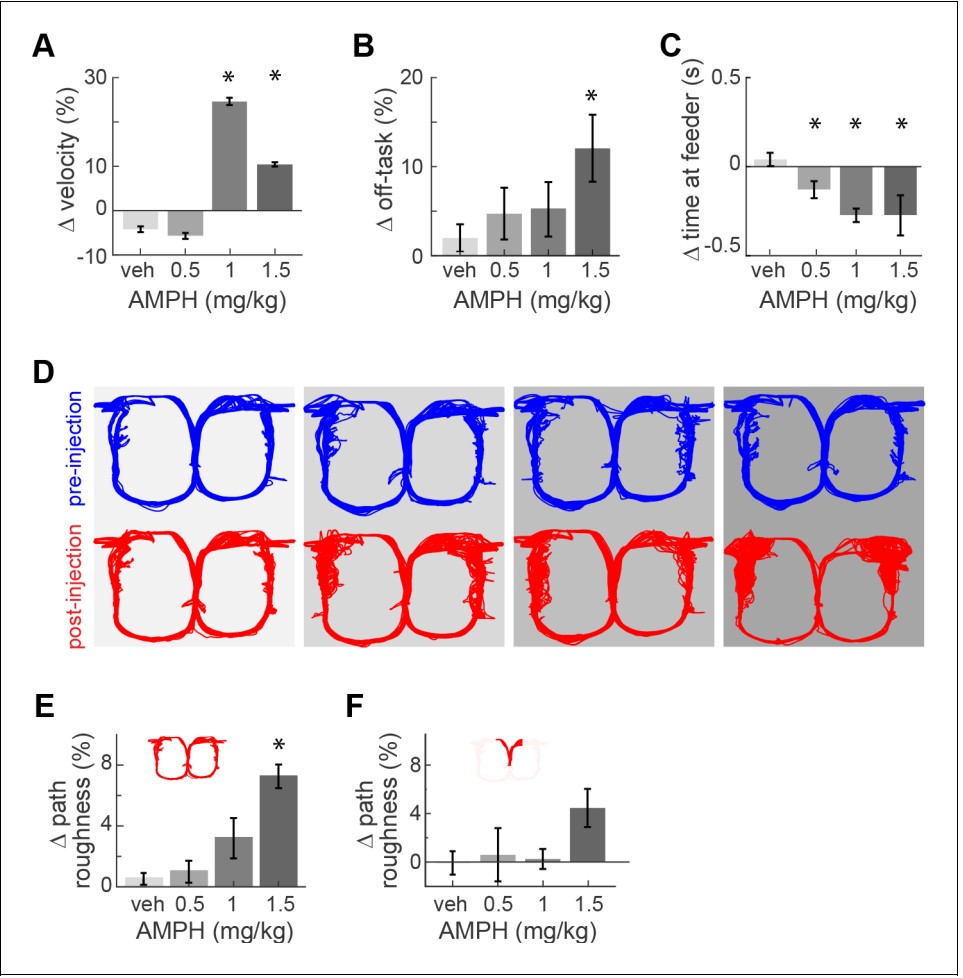

**Figure 2.** Task performance after AMPH injection. (**A**) Change of median running velocity after injection of AMPH with respect to values before injection. (**B**) Mean change in the relative proportion of trials with off-task behavior. (**C**) Change in the median time of occupancy at the start feeder. (**D**) Representative examples of running path superimposed for all trials in one session before (blue) or after (red) injection of vehicle or AMPH. (**E**) Mean change in running path roughness of all rats for the task epoch from the target feeders back to the start feeder. (**F**) Change in roughness of running path from the start feeder to target feeders, as measured by mean change in Hausdorff fractal dimension for all rats. Error bars show the Standard Error of the Mean (SEM) or Median (SEMd). Here and following figures: asterisks (*) indicate statistically significant differences at $\alpha$ = 0.05 with Bonferroni correction; only comparisons with saline are illustrated.

The online version of this article includes the following figure supplement(s) for figure 2:

**Figure supplement 1.** Time at target feeders decreases after AMPH.

AMPH administration also produced typical effects on reward-related behaviors. The rats' occupancy time at the reward feeders decreased with increasing dose (Kruskal-Wallis median test, $\chi^2(3)$ >136, p<0.0001 for all feeders). This occurred at the start feeder (*Figure 2C*) and target feeders (*Figure 2—figure supplement 1*). Such decreased feeder engagement after AMPH is consistent with previous reports (*Randrup et al., 1963*; *Randrup and Munkvad, 1967*; *Blundell et al., 1976*; *Blundell et al., 1979*; *Leibowitz et al., 1986*; *Floresco et al., 2008b*). In sum, moderate doses of AMPH in this task maintains animals' engagement in the task, even though they appear less motivated to consume the reward.

We next investigated effects of AMPH on running path. The reasons are twofold. First, ACC activity is highly sensitive to the running path, so any gross changes in path trajectory or variance may confound the decoding of other information (*Euston and McNaughton, 2006*). Second, it provides an additional indicator of task engagement. We therefore use the distribution of running path smoothness as an additional measure of task engagement and/or psychomotor effects. When analyzed over the entire track, the running path of rats became more variable after AMPH, and this effect increased as the dose of AMPH increased (*Figure 2E*; ANOVA, $F_{3,18}$ = 3.47; p=0.038; power = 0.77). This is particularly evident on the return arms of the track from the target feeders back to the starting feeder. The inter-trial path variance was not significantly affected in the segment from the starting feeder to target feeders (*Figure 2F*; ANOVA, $F_{3,18}$ = 1.83; p=0.178), although it trended to be higher at 1.5 mg/kg. These data indicate that task performance is not disrupted by lower doses, but begins to deteriorate at 1.5 mg/kg.

The behavioral data suggest that AMPH's effect in this study is typical of past studies, in which lower doses facilitate task engagement, but high doses disrupt it. Furthermore, measures of reward consumption decrease with increasing AMPH dose.

## ACC signals effort and reward

The ACC is well known to signal effort and reward. In order to visualize where in the present task ACC neurons signaled these variables, we segmented the track into 36 spatial bins and tested if the mean firing rate of each cell in each bin was significantly different during high-effort versus low-effort trials (t-test, p<0.05). Nearly 20% of cells signaled upcoming effort while on the approach to the barrier, and this proportion ramped up to nearly 35% during the climb/jump to the reward platform (*Figure 3A*, solid line). Conversely, about 10–15% of ACC cells in our sample encoded reward, and this proportion did not vary much during the task (*Figure 3B*, solid line). This is consistent with previous reports (*Cowen et al., 2012*). Administration of vehicle (saline) did not grossly affect the relative proportions of effort-related or reward-related cells (*Figure 3A–B*, dotted lines). This suggests that the relative proportion of cells signaling these task variables is consistent throughout the session. This is important because we use a within-session task design in order to observe AMPH-related

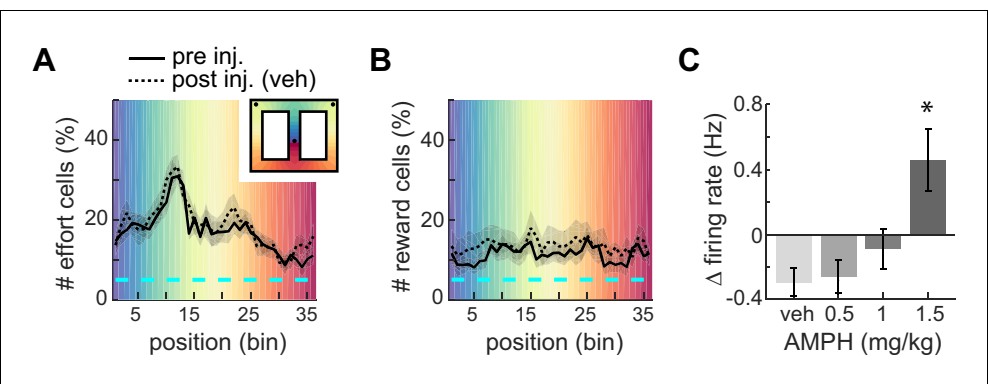

**Figure 3.** Proportion of ACC neurons encoding effort or reward. (**A**) Mean proportion of recorded neurons that discriminate barrier height in pre- or post-injection conditions. The proportion increases up to the barrier traverse. The shaded region surrounding the curves indicates SEM. Background colors correspond to each of the 36 spatial bins of the maze shown in the inset. (**B**) Mean proportion of recorded neurons responding to the reward. (**C**) Relative change in mean firing rate after injection (mean and SEM).

changes in signaling of the same set of neurons, and some neural correlates may change during the session. Indeed, we found that the mean firing rate is lower after vehicle administration (*Figure 3C*), which almost certainly reflects the typical decrease in firing rate as sessions progress. This decrease is attenuated as AMPH dose increases, and the firing rate is increased at the highest dose relative to the first half of the session (*Figure 3C*; $F_{3,1262}$ = 7.05; p=0.0001). This increase of ACC firing rate by AMPH is consistent with previous reports (*Lapish et al., 2015*), and again suggests that AMPH is having a typical effect in the present study.

## AMPH compresses the encoding of utility by single-units

The analysis of costs and benefits is typically formalized through the concept of utility, which can include many features (*Glimcher et al., 2009*). Here, we consider the joint encoding of effort and reward by individual ACC neurons. We use a linear regression approach, which is a standard method for discriminating when a continuous or binary predictor variable (reward or effort here) is informative of another variable (firing rate). We first computed the correlation of each neuron's firing rate with effort, and independently computed its correlation with reward volume, in each spatial bin from the starting feeder to the target feeders. We then analyzed the distribution of all cells in the effort-reward space for each spatial bin using principle component analysis (PCA). The first principle component (PC) reveals the primary axis of variance.

The distribution of points in the effort-reward space prior to drug reveals two interesting features (*Figure 4*). First, the distribution of points has a downward diagonal trend (i.e. the slope of the first PC is negative). This indicates that the joint encoding of reward and effort by individual ACC neurons is typically anti-correlated, which is expected of a utility signal. Further, the maximum variance of data is explained by neurons located in both the second ($Q_{II}$) and fourth ($Q_{IV}$) quadrants (Quadrants are numbered as in geometry, starting from the upper right corner and progressing in an anticlockwise direction). Neurons of quadrants $Q_{II}$ and $Q_{IV}$ have opposing signaling of utility. Neurons of $Q_{IV}$ tend to generate more action potentials for high-utility conditions (i.e. when the effort is low or the reward volume is large), and fire less in low-utility conditions (i.e. high-effort or small-reward). Neurons in $Q_{II}$ exhibit the inverse relationship among firing and value. The second revealing feature is that ACC cells are correlated with future reward and effort contingencies. This is evident by the non-uniform distribution of points in the spatial bin immediately following departure from the start feeder (*Figure 4A*, explained variance is 63, 62, 61, 61% for pre-inj, and 62, 63, 59, 57% for post-inj with increasing dose). This is nearly 100 cm in advance of the barrier and target feeder that are the basis of the correlation.

The slope of the first PC becomes steeper with increasing AMPH (*Figure 4A-B*), indicating a loss of neural firing correlation with reward. This relationship holds for the remainder of the spatial bins (*Figure 4—figure supplement 1*). Circular statistical analysis of the PCs from each spatial bin (first pooled over all sessions with similar treatment) reveals that the post-injection PCs are significantly more vertical than the pre-drug condition for AMPH, whereas saline injection has no effect (*Figure 4B*; Kuiper two-sample test, k = 160; p=0.028 for 0.5 mg/kg, k = 176; p=0.007 for 1 mg/kg, k = 240; p<0.001 for 1.5 mg/kg AMPH). To ensure that this is not an effect of heterogeneous sample sizes among conditions, we ran a bootstrap analysis (100 repetitions) in which we randomly sub-sampled the data to obtain the same number of neurons for each condition. The PCs were highly stable, and the results did not change with down sampling. Moreover, the percentage of total variance explained by the first PCs remains above 60% for all conditions, which further indicates that the results of the PCA are reliable. The rotation of neural tuning toward the vertical effort axis indicates that the encoding of reward is 'compressed' more than is the encoding of effort. Furthermore, the explained variance by the first principal component is significantly decreased at 1.0 and 1.5 mg/kg, which further indicates an overall loss of utility signaling at higher doses of AMPH (*Figure 4C.*; ANOVA $F_{3,60}$ = 17.3454; p=3 x $10^{-8}$). This effect might be through a breakdown in the encoding of effort or reward (e.g. points moving toward the origin), or an increased dispersion in effort-reward coding by shifting some neurons to the first quadrant in which the neurons respond positively to both effort and reward. In either case, these neurons fail to encode utility. Note that this analysis includes all neurons so as to eliminate any selection bias that may occur by first categorizing cells as utility or other classes based on statistical thresholds. We next analyze statistics of cells first identified as encoding reward or effort.

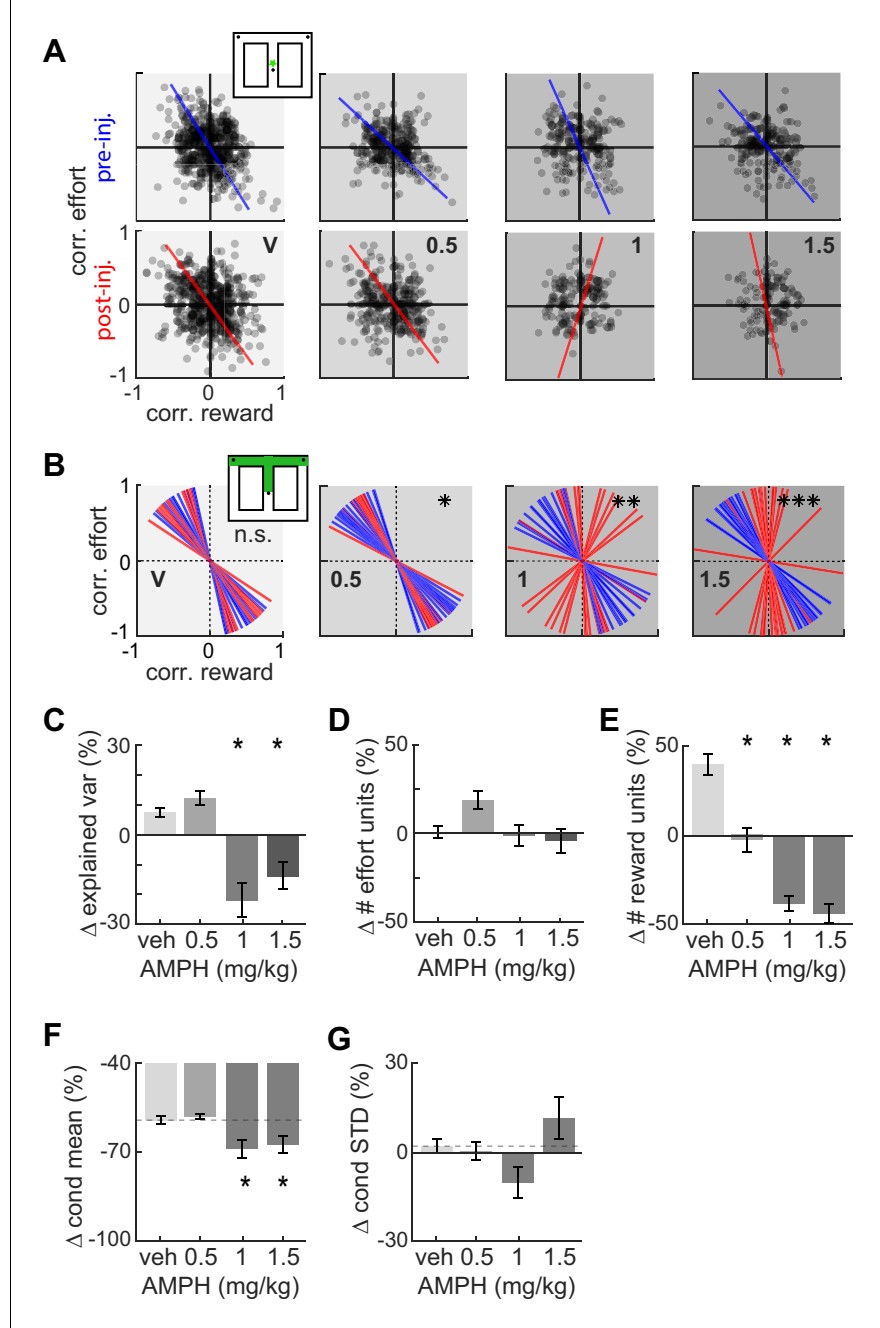

**Figure 4.** Effect of AMPH on the encoding of reward and effort by single neurons. (**A**) Joint effort-reward encoding by ACC neurons in all sessions in the spatial bin immediately following departure from the starting feeder (star in inset). Each black circle shows the Pearson correlation coefficient between the amount of effort and firing rate of one neuron, plotted against the correlation value for reward volume and firing rate of the same neuron. The lines indicate the first principal component (PC) coefficient during pre-injection (blue) and post-injection (red) for all units in the dataset recorded for each dose. (**B**) First PC of the effort vs reward correlation distribution for cells recorded in all sessions with similar treatment. Each line represents one spatial bin from center feeder to side feeder. Blue lines indicate PCs prior to injection, and red lines indicate the PC for the same cells after the injection. Statistical difference of the coefficient distributions pre versus post injection is computed by the Kuiper two-sample test, and indicated by asterisks: * significant at p<0.05, **: p<0.01, ***: p<0.001. (**C**) AMPH-evoked changes in mean explained variance by the first PC of effort-reward distributions, showing that the relationship between reward and effort for individual cells breaks down under higher AMPH doses. (**D**) Post-injection change of the proportion of effort-selective cells, indicating that effort signaling is not reduced by AMPH. *Figure 4 continued on next page*

*Figure 4 continued*

(E) Post-injection change in the proportion of reward-selective cells, showing that AMPH decreases reward encoding as measured by the number of cells discriminating reward volume. (F) Post-injection change in the absolute mean difference of firing rate of effort discriminating cells among trials of large vs small reward. The signaling of reward volume is decreased under 1.0 and 1.5 mg/kg. (G) AMPH-related change in the variance of neural firing around the mean firing rate in large and small reward (i.e. class-conditioned variance). The variance is not statistically different under AMPH, but tends to decrease under 1.0 mg/kg and increase under 1.5 mg/kg. Data show mean and SEM. Asterisks (*) indicate statistically significant differences at p<0.05; only comparisons with vehicle are illustrated.

The online version of this article includes the following figure supplement(s) for figure 4:

**Figure supplement 1.** Effort-reward tuning in key task epochs.

---

The preceding analysis suggests that neurons jointly encoding both effort and reward tend to lose sensitivity to reward more so than effort. We next sought to determine changes in the signaling of these variables independently, by computing the proportion of cells with significantly different firing rate for different levels of effort, or reward (ANOVA; p<0.05, and moderatley large effect size of partial $\eta^2$ >0.138). The proportion of units encoding effort shows a weak 'inverted U' shape with increasing AMPH, driven by the increase in the proportion of effort cells at 0.5 mg/kg (*Figure 4D*; ANOVA, $F_{3,60}$ = 3.73; p=0.016). No dose of AMPH caused a significantly different proportion than vehicle. Conversely, the proportion of units encoding reward strongly decreases with increasing AMPH (*Figure 4E*; ANOVA, $F_{3,60}$ = 46.04; p=14 x $10^{-16}$). In sum, high doses of AMPH reduce reward signaling in ACC, but do not appear to reduce signaling of effort. This is consistent with our behavioral observations that AMPH-treated rats appear less interested in consuming the reward. These results therefore suggest that reward is devalued by AMPH to a greater extent than effort.

The reduced number of reward signaling cells following AMPH could occur because the mean difference of firing rate in the two conditions diminishes (i.e. attenuated signal), or because the variation of firing increases (i.e. increased noise). We therefore computed signal amplitude and variation before and after AMPH, in all cells significantly discriminating reward value prior to drug. The difference of firing rates for large and small rewards decreased as AMPH increased (*Figure 4F*; $F_{3,1935}$ = 4.1; p=0.006). Note that the difference is negative after vehicle, likely reflecting reduced motivation for the reward as the animal becomes sated. AMPH appears to accelerate this process. Conversely, the standard deviation of firing from trial to trial was not significantly different under AMPH (*Figure 4G*; $F_{3,1935}$ = 2.06; p=0.104). Nonetheless, it shows a trend in which variability of firing is reduced by 1.0 mg/kg, but increased by 1.5 mg/kg. This is consistent with changes in ensemble variance shown later in this report. Note that the standard deviation is computed independently for small and large rewards before averaging, so the reduction is not a consequence of reduced difference of means among large and small reward trials. In sum, these data indicate that the loss of reward signaling for intermediate doses of AMPH (0.5 and 1.0 mg/kg) is primarily due to the signal reduction, rather than an increase in noise. At higher doses, both may play a role.

## Ensemble ACC activity encodes task epoch, task features, and past/present/future events

We next conducted a state-space analysis of simultaneously recorded neurons in order to assess how AMPH affects temporally evolving patterns of neural ensembles. We used a method termed Gaussian Process Factor Analysis (GPFA) to reduce the dimensionality of the data. This algorithm is particularly advantageous for producing smooth trajectories in low-dimensional space from high-dimensional processes with discrete events, such as action potentials (*Yu et al., 2009*). Similar to PCA, GPFA serves to capture as much variance as possible and does not optimize for any particular information present in the data (e.g. reward, location). In the 3D space of the first three GPFA factors, our ACC data form trajectories that move smoothly in the reduced space as the trial progresses (*Figure 5*). The trajectories discriminate task epochs, but are highly similar across trials of the same type. The trajectories diverge for trials of different effort or reward (*Figure 5*, right panel). For instance, the trajectory at the barrier climb deviates in high-effort trials as compared to low-effort ones. This is expected from the single-unit analysis, which showed that approximately 1/3 of ACC neurons discriminated effort during this epoch. Note that the ACC trajectory diverges widely on the

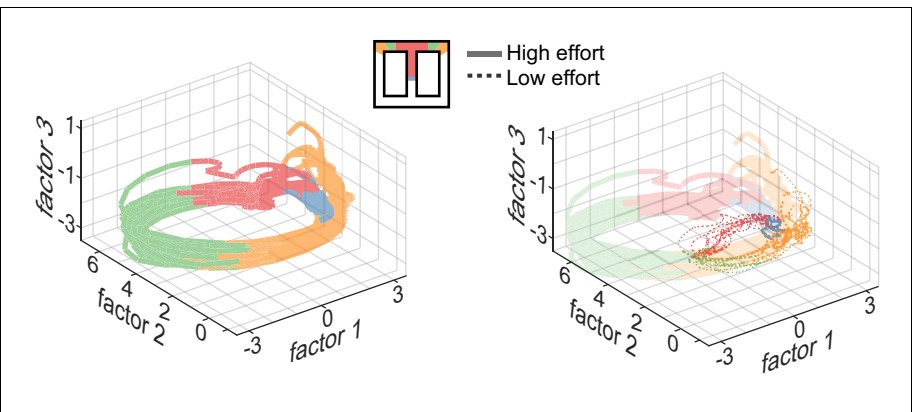

**Figure 5.** Neural trajectories encode task sequence. Plots show trajectories of neural encoding projected into low-dimensional space, and are color-coded by the sequential task epochs shown in the inset. Several trials are superimposed, revealing that the patterns of neural activity are distinct for each epoch but are highly similar across trials of the same type. The trajectories diverge for trials requiring different effort (right panel), and the divergence begins well before the actual barrier traverse, which occurs in the green region, as seen by divergence of encoding in the central stem of the track (red region).

approach to the turn (red shading), which is well before the barrier climb. This is consistent with the single unit analysis showing correlation of units on the middle segment, and further suggests that the ACC is sensitive to *upcoming* events.

The state-space analysis reveals another interesting novel phenomenon that cannot be discriminated directly from individual units. We analyzed trials in which the animal approached the barrier, then turned around and backtracked to the center feeder, and finally completed the trial by approaching and climbing the barrier. During this second approach and trial completion, the ACC diverged greatly from its typical trajectory during this task epoch, even though the path and velocity of the rat was very similar during both approaches (*Figure 6*). It thus appears that the ACC is sensitive to the recent history (or context) in which the task-related activity occurs. In sum, these data indicate that the trajectory of ACC activity in the reduced space is highly sensitive to the task epoch, task features (e.g. barrier climb), as well as the sequence of past and future events. Moreover, it shows that deviations from the trajectory are concomitant with off-task behavior. Changes in the trajectories, therefore, likely reveal changes in the neural processes directing task engagement and execution.

## AMPH contracts ACC state-space occupancy at low doses, and expands it at high doses

We next examined the effect of AMPH on neural trajectories. Low-dose AMPH caused a contraction of trajectories in state space for both large-reward trials (*Figure 7B*; ANOVA, $F_{3,371}$ = 28.28, p=2 x $10^{-16}$) and small-reward trials (*Figure 7C*; ANOVA, $F_{3,325}$ = 19.78, p=8 x $10^{-12}$). This is not likely explained by differences in running paths. This is because reduced state-space volumes suggest less variance of neural firing. This would require less variance in running paths, which does not occur (*Figure 2D-F*). It is also possible the first three latent factors capture less variance of the neural activity after AMPH, in which case the reduction in volume is likely an artifact of the methodology. We therefore compared the explained variance of the first three factors and found that it was not reduced by administration of 0.5–1.0 mg/kg AMPH (*Figure 7D*; confidence intervals do not deviate from 0). The contraction therefore appears to be a neural phenomenon, rather than a methodological artefact. As opposed to the reduction of state-space volume under low-dose AMPH, we instead found that 1.5 mg/kg caused an expansion (*Figure 7B-C*). Because the roughness of the running path increased at this dose, it is possible that these phenomena are related.

It has been suggested that the neuromodulators elevated by AMPH could increase the dynamical stability of cortical activity (*Durstewitz et al., 1999*; *Compte et al., 2000*; *Durstewitz et al., 2000*; *Brunel and Wang, 2001*; *Gruber et al., 2006*; *Cano-Colino et al., 2013*). If so, we would expect a reduction in the variation of neural trajectories across trials of the same type. We found that the

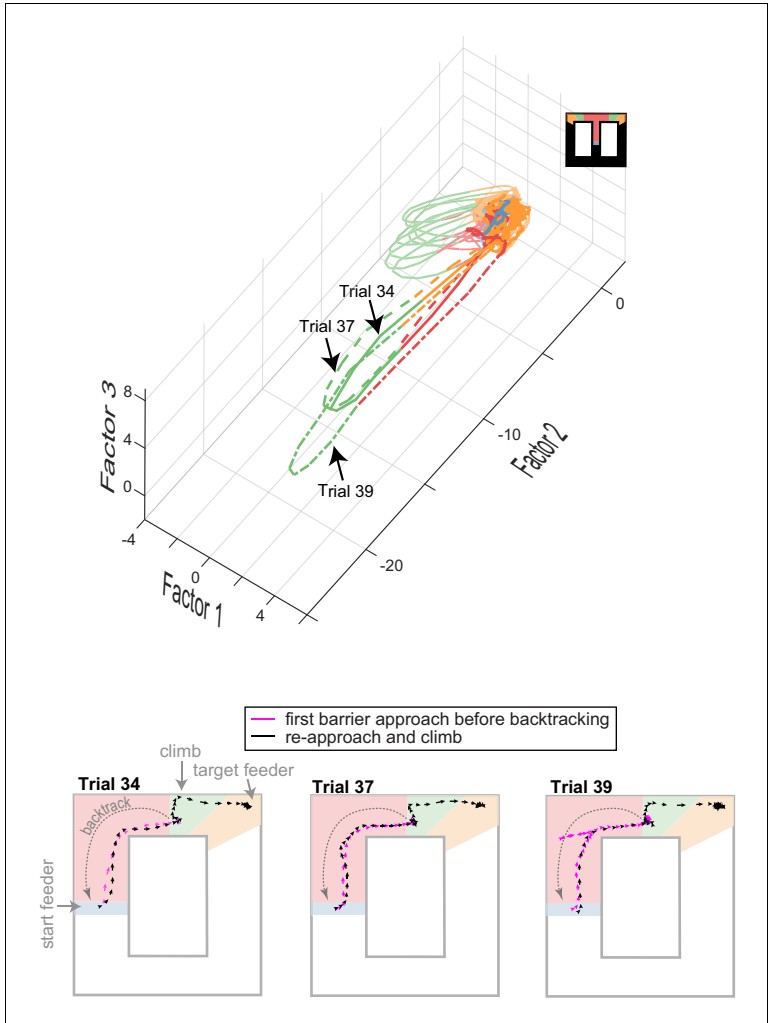

**Figure 6.** Ensemble encoding is task-specific. Top plot shows ACC trajectories of population activity in low-dimensional space for several trials in one session, including three trials with off-task behavior (identified by trial numbers). In these cases, the animal approached the barrier, then backtracked from the barrier to the start feeder before returning to commit the climb and reach the target feeder. The running paths and velocity vectors are shown in the lower panels. Note that the neural trajectories deviate largely when the animal is on the second approach and climb (dashed lines in upper panel) as compared to trials with no off-task behavior (solid red-green-orange sequence of lines). This deviation occurs, even though the paths and velocity of the initial approach (fuchsia) and re-approach (black) are similar. This suggests that ACC is sensitive to past events – in this case, off-task behavior.

mean variation of trajectories showed a trend to vary under AMPH but were not significantly different with the present sample (ANOVA; large-rew: $F_{3,62} = 2.11$; p=0.108, and small-reward: $F_{3,62} = 2.37$; p=0.079). Nonetheless, the variation tended to decreases relative to baseline following 0.5 and 1.0 mg/kg, but then tended to increase under 1.5 mg/kg, for both small reward trials and large reward trials (*Figure 7E–F*). The pattern of reduced variation under low AMPH and increased variation under high AMPH is generally consistent across task epochs but is particularly evident when animals were at the target feeder (*Figure 7*.G). The pattern of dose-dependent modulation of variance described above is statistically significant at the target feeder (ANOVA; $F_{3,62} = 3.93$, p=0.012). These data support the hypothesis that neural dynamics are stabilized by low-to-moderate AMPH, but become unstable under high doses.

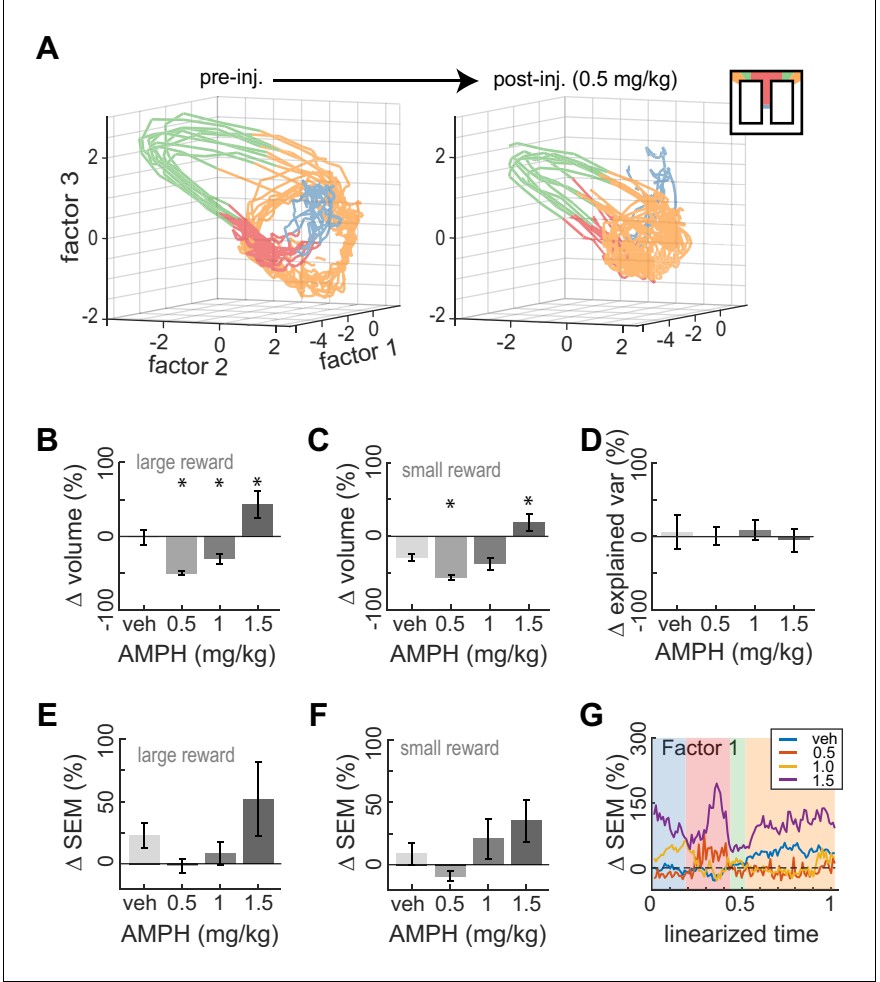

**Figure 7.** Dose-dependent contraction of neural trajectories. (**A**) Example of ACC neural trajectories before injection (left panel), and after injection of 0.5 mg/kg AMPH (right panel) plotted into the same low-dimensional space for one session. The volume enclosed by the envelope (outer hull) of the trajectories contracts after AMPH. (**B**) Post-AMPH changes in mean trajectory volume over all sessions and trials with large reward. The volume decreases for intermediate doses, and expands at the highest dose. (**C**) The same dose-dependent volume changes occur for small-reward trials. (**D**) Changes in the amount of variance explained by the first three factors after AMPH. Medians and 95% confidence intervals are shown, revealing that explained variance is not significantly reduced after AMPH because the confidence intervals encompass 0. (**E**) Change in SEM of neural trajectories after AMPH on large reward trials averaged over all the first three factors and all spatial bins, showing a reduction in trial-by-trial variance for low-dose AMPH. (**F**) Same as panel E for small reward trials. (**G**) Change in SEM for the first factor across task epochs. Data are averaged over all sessions with similar treatment for large reward trials. Asterisks (*) indicate statistically significant differences at $p < 0.05$; only comparisons with vehicle are illustrated. Panels B, C, E, and F show means and SEM.

## Discussion

We investigated the effects of systemic AMPH administration on ACC encoding and dynamics in rats performing a task with variable effort and reward. Our aim in this study was to use neural recordings to better understand why AMPH typically causes animals to become more engaged in tasks and to increase effort for food rewards, while also decreasing their motivation to consume the reward. We observed these typical behavioral features in the present study. Rats ran faster, but spent less time consuming rewards as AMPH increased. We observed several neural correlates of task performance that were sensitive to AMPH. Notably, many individual ACC neurons jointly encoded effort and reward, which is expected of cells involved in computing utility. Many units signaled positive utility; they positively correlated with the reward amount and negatively with the

effort level. AMPH injection increased the dispersion of effort-reward coding in the population, but the rotation of the principle axis of the population toward the effort axis reveals that the suppression of reward signaling by single units is the primary effect. This effect was mirrored by all neurons encoding reward or effort. AMPH injection thus decreased reward signaling in ACC while having little effect on effort signaling, and this coincided with reduced time at the feeder. Note that the reduced reward signaling emerged well before rats reached the target feeder, and so is indicative of altered anticipation, rather than changes in sensory aspects of reward consumption. Therefore, it is possible that the reduced reward signaling is a causal factor in the reduced motivation for reward that is a hallmark of AMPH across species (*Blundell et al., 1976*; *Blundell et al., 1979*; *Leibowitz et al., 1986*).

The state-space analysis of ensemble activity (mean of 55 ± 6.7 putative pyramidal neurons per session) revealed smoothly varying trajectories specific to task epochs and conditions. These trajectories were highly similar across trials of the same type, suggesting they reflect task-specific information. Besides encoding the present state of the animal on the track, the trajectories also reflected future events. This is evident in the deviation of trajectories prior to turning to reach the barrier in high vs low effort trials and is consistent with our finding that many single units were correlated with future reward and effort. Trajectories were also sensitive to past events. In particular, the trajectories deviated extensively following off-task behaviors, even as the rats re-engaged in the task and performed in a typical manner. Thus, the trajectories are task-specific, and sensitive to past, present, and future states/actions. AMPH had dose-dependent effects on the trajectories. Their modulation during the task (volume in state space) and trial-by-trial variance contracted under low-doses, but expanded under high doses. This coincided with increased task engagement at the lower doses, but frequent disengagement at high doses. Note that we excluded trials with off-task behavior from the neural analysis of volume, suggesting that the off-task behavior is triggered by processes related to the expansion of state-space occupancy and/or variance, rather than the inverse relationship.

AMPH increases extracellular concentrations of dopamine, norepinephrine, and other neuromodulators in the prefrontal cortex and other regions (*Chiueh and Moore, 1973*; *Pum et al., 2007*). The idea that such neuromodulators affect brain dynamics in the neocortex by altering the stability of attractor states has a long history in computational studies (*Durstewitz et al., 1999*; *Compte et al., 2000*; *Durstewitz et al., 2000*; *Brunel and Wang, 2001*; *Gruber et al., 2006*; *Cano-Colino et al., 2013*). Attractors are stable states of activity produced by particular configurations of synaptic connectivity and neural excitability. Stability in this context indicates that the network resists small perturbations (inputs that drive ensemble activity to patterns different than the attractor state) so as to re-settle in the stable pattern. Increased stability means that the state can resist larger perturbations. Attractors can take different forms. The simplest is point attractors, which form a static pattern of activity in time. Another form is line attractors, in which activity propagates smoothly along a 'valley' of stable points, but resists perturbations that push the system up the 'hills' on either side. Computational studies have predicted that dopamine can stabilize these types of attractors as well (*Gruber et al., 2006*).

Recently, Lapish and colleagues tested the hypothesis that low-dose AMPH stabilizes attractor states in dorsal medial prefrontal cortex, whereas high doses destabilizes them, during a working memory task (*Lapish et al., 2015*). They found that relative to saline, 1.0 mg/kg AMPH reduced variation of ensemble states after dimensionality reduction, whereas 3.3 mg/kg expanded it. This is consistent with the dose-dependent effects on state-space variation we report here (*Figure 7*). Interestingly, these authors report that neural states tend to form distinct clusters separated in space for different task epochs (*Balaguer-Ballester et al., 2011*; *Lapish et al., 2015*). In contrast, we found that ACC states do not cluster, but rather form smooth trajectories that evolve as the trial progresses. This difference may be due to the difference in dimensionality reduction methodology, task demands, and/or the number of units recorded. Here, we have a larger number of neurons in each ensemble, but use a dimensionality reduction method that is not optimized for anything other than capturing maximal variance in each successive factor, similar to principle component analysis (PCA). Nonetheless, the low-dimensional representation was highly specific for task epoch and task condition (effort, etc.) and was remarkably consistent across trials, suggesting that it does capture meaningful variance. Furthermore, it was sensitive to future events, as well as past off-task behavior. Based on these properties, we suggest that ACC may be executing some form of a continuous mental 'script' for task events and execution, which includes not only the present state, but also what is

about to happen. By 'continuous', we mean that the ACC is not only encoding information about the sequence of events, but also their relative proximity in time or space. This is consistent with the smooth ramping up of the proportion of effort-related cells as rats approach the barrier (*Figure 3A*). It is also consistent with the ramping up of primate ACC activity in anticipation of stimuli and reward (*Amiez et al., 2006*).

The contraction of state space occupancy and trend of smaller variance under 0.5 and 1.0 mg/kg AMPH may reflect increased stability of a line attractor. Further, we propose that the increased excitability under AMPH (*Figure 3C*; *Lapish et al., 2015*) could account for the increased speed of task execution. That is, it propels the activity state along the valley of the line attractor more quickly. Increased excitability can cause intrinsic perturbations that destabilize attractor states, but we speculate that the increased dynamical stability by AMPH counteracts this effect. We also speculate that the further increased excitability at higher doses may exceed the increased stability. This would result in higher variance and state-space occupancy, and more frequently generate perturbations sufficiently large to escape the attractor basin, so that the task-related trajectory cannot recover. The result would be a break in the behavioral script and frequent interruptions of task performance. These predictions are consistent with the neural and behavioral data observed here and elsewhere, as described next.

Our data provide a possible explanation for the apparently paradoxical effect of AMPH on task engagement and reward valuation. The reduced motivation for reward may follow from the reduced reward encoding. They continue to engage in the task, however, because of the increased stability of task-related neural trajectories. In short, the stability is sufficiently strong that it 'captures' the neural dynamics to an extent that it is more difficult for the brain to shift to other behavioural scripts. The result is that rats run through the behavioural script of the task, but are then not motivated to consume the reward. This provides an explanatory mechanism for many of the known effects of AMPH. As described above, AMPH has been shown to decrease feeding (*Foltin, 2001*; *Shoblock et al., 2003*; *Cannon et al., 2004*; *Wellman et al., 2009*) and decrease the sensitivity to reward omission (*Wong et al., 2017*), suggesting a devaluation of food rewards. Again, this may follow from the attenuation of reward encoding. Low doses of AMPH increase task engagement and execution speed in a variety of tasks (*Wilkinson et al., 1993*; *Foltin, 2001*; *Odum and Shahan, 2004*). This can be explained as a consequence of increased excitability balanced by increased stability. Similarly, it provides a possible explanation for the ability of low-dose AMPH to enhance attention, vigilance, and working memory in a variety of task settings and species (*Sostek et al., 1980*; *Ridley et al., 1982*; *Koelega, 1993*; *Solanto, 1998*; *Grilly, 2000*; *Shoblock et al., 2003*; *Silber et al., 2006*; *Sagvolden, 2011*). Specifically, these can be described as increasing robustness against perturbations unrelated to the task (*Gruber et al., 2006*). AMPH also consistently increases motoric activity (*Randrup and Munkvad, 1967*; *Groves and Rebec, 1976*; *Robinson and Berridge, 1993*; *Wilkinson et al., 1993*; *Sams-Dodd, 1998*), an effect we observed here. Our explanation is that it is a consequence of increased neural excitability constrained by increased attractor stability. This is not likely an effect mediated largely by ACC, because ACC-lesioned animals have little deficits in engaging motoric output (*Holec et al., 2014*; *Brockett et al., 2020*). This suggests that AMPH acting in other brain regions also influences task performance in our animals.

The control of behavioral output depends on sets of interconnected brain structures (*Balleine and O'Doherty, 2010*; *Gruber and McDonald, 2012*), and AMPH influences processing in several of them (*Chiueh and Moore, 1973*; *Pum et al., 2007*). Therefore, we cannot infer from our data where in the brain AMPH may be affecting task-related processing. Dopaminergic tone in the nucleus accumbens appears to be critical for effort-based decision-making (*Salamone et al., 1994*), but it is less clear if dopamine in the ACC also is needed. *Walton et al., 2005* reported that depleting dopamine innervation with 6-OHDA in the ACC did not alter the likelihood of effort-based responses, whereas *Schweimer and Hauber, 2005* found that such lesions reduced this type of responding. This discrepancy may be related to methodological differences between the two studies. Because the mPFC and striatum form functional circuits with other structures (*Middleton and Strick, 1997*; *Voorn et al., 2004*), we can use neural activity in ACC as a window into network processing. Previous studies have reported ACC activity in anticipation of effort and reward (*Sul et al., 2010*; *Cowen et al., 2012*; *Hashemniayetorshizi et al., 2015*). Our results show that ACC neurons primarily encode utility in the portion of the maze from the starting feeder to target feeders. It is likely that the animals anticipate the effort and reward of the upcoming trial because of the task

design; the animal was directed to alternate between left and right options, and the effort and rewards were invariant over blocks of 20 trials. Indeed, we found that many neurons encoded future utility when the animal had just left the starting feeder, and ensemble dynamics deviate among trials of high/low effort at this point. It therefore seems that animals track upcoming effort and reward, even though they are not able to choose among the target feeders. Their choice in the present experiment is whether to engage in the task or not. Our data are therefore more revealing of mechanisms of task engagement than choice. Indeed, the present data do not provide conclusive evidence as to why animals shift their preference to high-effort high-reward options under low-dose AMPH (*Floresco et al., 2008b*; *Bardgett et al., 2009*). Effort encoding was not affected by AMPH, so it does not appear that the brain discounts effort or cannot discriminate future effort levels.

One caveat to our study is that we used repeated injections of AMPH over several days to provide escalating doses in the same set of animals. We used this design because (1) drug naive rats will often not perform tasks if their first exposure is a high dose, and (2) the expense and effort needed to record neural ensembles prohibits us from collecting only one dose per rat. It is likely that AMPH had lasting effects on the brain. Repeated administration of high-doses AMPH (5 weeks of escalating daily dose up to 8 mg/kg) changes synaptic and dendritic density in the mPFC of untrained rats (*Robinson and Kolb, 1997*). Furthermore, withdrawal from methamphetamine following weeks of repeated exposure has been shown to reduce the preference of rats for high-effort, high-reward options (*Hart et al., 2018*). Our animals had far lower exposure (3–5 injections of 0.5–1.5 mg/kg). Nonetheless, we attempted to minimize confounds of AMPH-driven changes. We used an escalating dose to minimize total prior exposure on each recording session. We also intermixed control sessions (saline injections) between the days of AMPH. Lastly, we used within session contrasts prior to between-session comparisons. These latter two controls serve to capture accumulating changes in signaling that may occur as the experiment progressed. Furthermore, we used a before/after design in which the drug was always given in the second half of the experiment. This allowed us to observe the effect of AMPH on signaling of an identified set of cells. However, any consistent changes that occur during the session, such as satiation, may therefore confound the results. The between-session contrasts help mitigate this confound. The pattern of results we observed, particularly the fact that responses often had different polarity for high vs low doses, suggests that this potential confound is not strongly influencing our results or interpretations.

In conclusion, the data in the present study suggest that AMPH decreases reward signaling in circuits involving ACC, which may play a role in the well-known effect of this drug to reduce reward consumption behaviours. Further, the data support previous proposals that AMPH has a dose-dependent effect on the stability of neural dynamics in the prefrontal cortex, which could explain increased task engagement at low doses, but increased task disruption at high doses. Future studies are needed to identify which of the several neuromodulatory systems affected by AMPH play a role in these effects.

## Materials and methods

### Subjects and surgical procedure

Four adult male Fischer Brown Norway (FBN) hybrid aged 6 to 10 months were used in this study. Rats were born and raised on-site, housed individually in a 12 h-12h reverse light cycle room, and habituated to handling for 2 weeks prior to surgery. The fabrication and surgical implantation of head-mounted electrode drives was completed as previously described (*Euston and McNaughton, 2006*). Briefly, surgeries were carried out prior to any training. Animals were deeply anesthetized with isoflurane throughout the procedure (1–1.5% by volume in oxygen at a flow rate of 1.5 L/min). Each animal was implanted with a 'hyperdrive' consisting of 12 independently-movable tetrodes and two reference electrodes (*McNaughton et al., 1983*; *Wilson and McNaughton, 1993*). The hyperdrive bundle was centered at 3.00 mm AP, and 1.3 mm ML of left mPFC and angled 9.5 degrees toward the midline. A craniotomy was made around the electrode exit site of the drive, and the drive bundle was lowered to the brain surface. The dura was retracted, and the hyperdrive body was secured to the skull with anchor screws embedded in dental acrylic (Lang Dental, Wheeling, US). Following surgery, rats were given daily injections of 1 mg/kg meloxicam (3 days) and 10 mg/kg enrofloxacin (5 days). Tetrodes were lowered 950 μm from the skull surface after the surgery and then

gradually lowered daily over the next 2–3 weeks to reach the target depth. Food restriction began after the animal recovered from the surgery (7 days), and the amount of food given was adjusted to ensure that the animal's weight was at least 85% of the free-feeding weight for the duration of the experiment. The experiments were performed in dim light during the animal's waking phase. All procedures were performed in accordance with the Canadian Council of Animal Care and the Animal Welfare Committee at the University of Lethbridge.

## Experiment

The behavioral apparatus and data collection methodology have been described previously (*Mashhoori et al., 2018*). Briefly, we used an automated figure 8 maze (*Figure 1A*), which is a modified version of the classic T-maze frequently used in studies of effort-reward decision-making (*Salamone et al., 1994*; *Walton et al., 2002*). The track of the maze was 15 cm wide, and configured into a rectangular pathway measuring 102 × 114 cm. The maze contained a 'start' feeder on the central T-stem, where trials were initiated. One target feeder was located on each of two platforms located in the upper corners of the maze. The platforms could be elevated by a motor in order to require rats to climb or jump a vertical mesh to reach the platform containing the feeder. Animals descended from the platform by a ramp to return to the starting feeder. The elevation of the platform was 0 cm for the low-effort condition (level with the track) and was 23 cm for the high-effort condition. The reward was a high-calorie liquid (chocolate flavored Ensure). The volume was 0.03 ml for small rewards, and 0.12 ml for large rewards. The same small volume of reward was delivered at the center feeder in all trials so as to motivate the rat to return to the start position. Four impassable gates were located on the track, and served to restrict access to particular regions of the maze. One set of two gates prevented rats from traversing the maze in the backward direction once they were at the starting position. A second set of gates limited access to the choice feeders, and could force the selection of one target feeder. For the first week of training on the maze, both gates were open, so the animals could explore either side. Over the following 2–3 weeks, animals were trained on the maze with a mixture of forced trials (one gate to the platforms open) and free trials (both gates open). In these sessions, rats were exposed to all possible combinations of barrier height (2 levels of effort), reward volumes (two levels), and choice direction (two options). The barrier height was gradually increased during training (7.5, 15, 23 cm) to reach the target level (23 cm). Animals were considered well-trained when they chose the large-reward option (on free trials) more than 80% of the time. Rats then performed sessions to generate the data presented here. Because we planned a before/after drug design within each session, we reduced the number of task conditions so that we could collect a sufficient number of trials in each condition to estimate changes in behavior and neural encoding. To achieve this, we used only forced trials and organized the task into blocks of 20 trials in which the two options had either equal effort-reward contingencies (block 1), or varied either by reward volume (blocks 2 and 3, counterbalanced over right/left target feeders), or by barrier height (blocks 4 and 5, counterbalanced over right/left target feeders). Trials alternated between left and right throughout the session. Because direction (right/left target feeder) had little effect on neural signaling of effort or reward, we collapsed data over direction in the analysis. One animal performed the task with three levels of barrier, but we included data from only the highest level relative to the 0-height level.

Animals performed five blocks of trials (100 trials), and then received an i.p. injection of either vehicle (saline) or AMPH. They remained on the maze for 10 min with all gates closed to give time for the drug to take effect before resuming trials. Rats then repeated the same five block sequence. Rats quickly sensitize to AMPH, and withdrawal from repeated administration of high doses can affect effort-reward behavior (*Hart et al., 2018*). We thus employed four experiment design elements in attempt to control for the potential confound of sensitization and/or withdrawal: (1) we used a within-session design so that the pre-drug blocks serve as a control for lasting changes due to prior drug administration; (2) sessions were separated by at least 24 hr to provide some recovery time; (3) saline control sessions were collected on days between AMPH sessions so as to include some of any lasting changes from repeated AMPH administration; and (4) we used an escalating drug schedule, rather than counterbalancing dose order, for most animals. Three doses of AMPH (0.5, 1, 1.5 mg/kg) were used in this study. Only one dose (or vehicle) was given per daily session. The order of drug/saline injection across sessions was: vehicle, AMPH 0.5 mg/kg, AMPH 1 mg/kg, vehicle, AMPH 1.5 mg/kg. One rat received vehicle, AMPH 0.5 mg/kg, AMPH 1.5 mg/kg, vehicle,

AMPH 1 mg/kg, vehicle, AMPH 0.5 mg/kg. The rationale for this schedule is twofold. First, we have observed that rats often will not perform operant tasks if their first exposure to AMPH is a high dose. Secondly, escalating doses minimize the cumulative amount of prior AMPH exposure on each testing day, which likely minimizes the amount of neural adaptation. The block order is the same in each session, and the drug is always in the second half of the session. The potential confound of declining motivation within sessions is taken into account by computing dose-response curves across sessions (after within-session normalization).

## Histology

Following completion of the study, the recoding sites were marked by passing 10 µA direct current for 10 s through one electrode of each tetrode. Two or 3 days later, rats received lethal injections of sodium pentobarbital (100 mg/kg i.p.) and were perfused with phosphate buffered saline (PBS) and 4% paraformaldehyde (PFA). The brains were post-fixed for 24 hr in 4% PFA and then transferred and stored in 30% sucrose and PBS solution with sodium azide (0.02%). After fixation, the brains were sectioned coronally (40 µm), mounted on glass microscope slides, and labeled with cresyl violet. Digital images of brain sections were produced with a Nano-Zoomer slide scanner (Hamamatsu, Japan) and visually inspected to determine the location of marking lesions. Electrode endpoints were identified for three brains.

## Analysis

A total of 22 sessions were included in the analysis. The data include at least two sessions with saline treatment and one session of each drug dosage for each animal. Each session is partitioned into two phases, pre-injection and post-injection. The analysis contrasts the relative change in the post-injection phase with respect to the pre-injection phase. Sessions in which saline was injected serve as the control for features of signaling that change as a function of task sequence (e.g. block order, satiety).

### Preprocessing

Camera-based tracking of LEDs located on preamplifiers attached to the recording drives were used to estimate the position of animals on the maze. We used image registration methods in MATLAB to correct any maze or camera shifts or rotations between the recorded sessions in order to normalize maze position across sessions.

Recorded spikes waveforms were first automatically clustered using KlustaKwik (author: K. D. Harris, Rutgers-Newark) and then manually sorted using MClust (David Redish, University of Minnesota, Minneapolis). Putative individual neurons were then manually classified as either pyramidal neurons or interneurons based on established methods and criteria (*Barthó et al., 2004*). Neural analysis was restricted to only putative pyramidal neurons. In total, 1266 cells were recorded, of which 1209 met the criteria for pyramidal neurons.

### Behavioral analysis

The effect of AMPH on task performance was investigated using four behavioral measures: the variance (roughness) of rat locomotion trajectories; the median locomotion velocity; the number of off-task episodes (grooming, rearing, reversing running direction); and time spent at feeders consuming reward. First, the roughness was quantified by the Hausdorff fractal dimension (*Hausdorff, 1919*; *Gneiting et al., 2012*), which in our case increases as the variance (in time) of an animal's running path increases. We first superimposed all paths before injection (trials 1–100) at the resolution of the video camera (2.36 × 2.36 mm), and produced a binary image of pixels that were traversed on any trial and those that were not. The Hausdorff dimension was computed on this matrix. The procedure was repeated for trials after injection (trials 101–200). Second, the running velocity was calculated by the change in rat's position (in pixels) between every 111 ms time bin (one pixel/sec is equivalent to 2.36 mm/sec). Third, trials with off-task behaviors were identified by manually scoring the video. Off-task behaviors included long pauses on the track, continuous repetition of circling, backtracking, or grooming. Since these stereotyped behaviours are known to be induced by AMPH (*Randrup et al., 1963*; *Randrup and Munkvad, 1967*), we expected to observe them more often in the post-injection phase. Finally, the reward consumption time was measured by the duration that the animal's

velocity dropped below an empirically selected threshold (59 mm/s at the central feeder, and 118 mm/s at target feeders) at each of the three reward zones. The velocity threshold is not closer to 0 mm/sec because rats perform head-bobs and other small (but fast) movements during reward consumption. The relative change in each of these four measures was found with respect to the pre-injection phase, producing one value per session. The marginal measures of these changes were then compared between vehicle and drug sessions.

For normally distributed data, we computed means and used ANOVA to detect significant differences of marginal means. We report corrected values (Greenhouse-Geisser) for data with skewed variance, but otherwise approximate the normal distribution. For data with distributions inappropriate for parametric tests, we used Kruskal-Wallis for testing difference of medians. Longitudinal experimental designs in which individual subjects contribute multiple data points are often analyzed with a random factor (e.g. repeated measures) to correct for the high covariance of data points from each subject. Here, we instead correct the covariance by computing within-session differences prior to ANOVA. We computed statistical power of inferential statistical tests using G* Power (*Faul et al., 2007*) for any test near the threshold of $\alpha < 0.05$. Bonferroni correction is used in all post-hoc comparison of individual means to correct for multiple comparisons.

## Single unit encoding of effort/reward

We first linearized the track by dividing each loop of the figure 8 maze into 36 rectangular bins, starting at the central feeder (*Figure 3-inset*). The firing rate of individual neurons was computed as the number of spikes in a 0.3 s time bin centered on each of these spatial bins. Besides drug, three task parameters were controlled during sessions: reward volume; barrier height; and the location of the reward delivery (right/left target feeder). We determined if a unit encoded any of these features by computing its marginal means (for each spatial bin and trial) among blocks in which only the corresponding parameter was different. For example, we computed conditional means for high effort and low effort from trials in which reward size and reward location were the same, but effort was different (blocks 4–5). A neuron was considered to be responsive to a feature if it significantly discriminated low- and high-conditions on either the left or right side (ANOVA, $p < 0.05$, and moderately large effect size of partial $\eta^2 > 0.138$). The percentage of effort-responsive neurons was found in each of the 36 spatial bins after removing the cells with no discrimination of effort, reward, or feeder location from the population. Grouping the sessions with similar treatments, the average portion of responsive neurons were obtained for pre- and post-injection phases separately. The same procedure was followed to compute responsiveness to reward. This procedure commits multiple comparisons in that all neurons are re-tested in each spatial bin, and therefore is expected to increase type I (false positive) errors. We do not correct for multiple comparisons in this particular analysis because (*i*) it follows standard practice, and (*ii*) we are computing within-session differences on the same cells over the same task conditions (before/after drug). Because the false discovery rate should be similar in both conditions, the comparison between conditions should be relatively invariant to errors of multiple comparison.

The main aim of this study was to investigate the effect of AMPH on the task-related encoding of effort and reward. We first investigated signaling properties of single units by the distribution of their encoding of reward and effort. We use an approach following linear regression analysis, which is the standard method for detecting the relationship between binary predictor variables (reward and effort in the present case) and a continuous variable (firing rate) (*Freedman, 2009*). Encoding strength was computed by the Pearson correlation coefficient between the spiking rate of each neuron and the two levels of effort (or reward) in each of the first 16 spatiotemporal bins (from the central feeder to the bin after the target feeder). Thus, each neuron had a correlation coefficient for reward and for effort. Next, we quantified the distribution of neurons in the effort-reward (E-R) space using principal component analysis (PCA). We used the first Principle Component (PC) to determine the axis of maximum variability of the aggregated population of cells (from all sessions) in the E-R space for each spatial bin. To account for the different numbers of cells recorded at each drug dose, we bootstrapped data by randomly selecting the same number of neurons for pre- and post-injection phases and repeated this process 100 times per spatial bin. Next, we statistically tested the difference of the distributions of principal component coefficients of the two phases for each treatment by the Kuiper two-sample test, which is the circular analogue of the Kolmogorov-Smirnov test. Lastly,

we computed the amount of variance explained by the first principal component for pre- and post-injection phases to assess whether the relationship was rotated or began to break down. This is again based on 100-times bootstrapped data selection to compensate for unequal number of samples. Note that this analysis does not use any selection criteria, but instead utilizes all recorded cells. Moreover, the test does not depend on assumptions of variance (sphericity, etc.) because it is not parametric, but rather utilizes the sampled distributions.

### Analysis of ACC population activity

The amount of time required for rats to locomote from the starting feeder to a target feeder varies from trial to trial. We therefore mapped the neural activity from each trial to a common linear representation for each of four key task epochs: reward-consumption at the starting feeder; interval from the starting feeder to the barrier; climbing the barrier; and reward-consumption at the target feeder. In other words, spike times from each trial were linearly expanded or contracted to span a 'reference' epoch. The reference was obtained by calculating the average occupancy of all animals in each spatial bin in each epoch (prior to injection). This reference is then used for all animals. Mapping each trial to the reference aligns the activity so as to facilitate the detection of treatment effects and allow comparisons between sessions.

In order to investigate ensemble ACC population activity, we first used Gaussian Process Factor Analysis (GPFA) to reduce the dimensionality of the neural data (*Yu et al., 2009*). GPFA extracts the latent structure embedded in temporally evolving population activity. Its key advantage over other dimensionality reduction techniques (e.g. PCA) is that it optimizes the width of the smoothing kernel needed to convert discrete events such as action potentials to a continuous signal. It produces neural trajectories, which are the time series of mappings into the low-dimensional space. Each point in the low dimensional space represents a linear combination of activity from multiple neurons. In each experimental session, the eight most important latent factors (akin to principle components) were extracted from the pre-injection data. These factors represent the directions in the high-dimensional space along which projections show largest variance. Factors 1–3 captured most of the variance, and so we focused analysis on these first three. We removed trials in which the animal performed abnormally slowly (>20 s from starting feeder to target feeder) or had off-task behaviour. The post-injection trajectories were projected onto the same low-dimension space that was computed from the pre-injection activity data, which allows us to directly compare trajectories before and after AMPH. That is, all trajectories from the same session are mapped into the same latent space.

We next sought to determine if AMPH affected the fidelity of ensemble ACC encoding. We first computed the state-space occupancy of neural trajectories in 3D space made by the first three GPFA factors. The boundary of the 3D space was obtained by the convex hull of the trajectory from the central to the target feeders; the enclosed volume was calculated by summing over segments of the hull determined via triangulation. We next computed the change in volume post injection independently for large reward trials and low reward trials, and tested if the change in volume after AMPH was significant across sessions by ANOVA.

## Acknowledgements

This work was supported by the National Sciences and Engineering Research Council of Canada (NSERC), Alberta Innovates Health Solutions, and the Beswick Fellowship (SH).

## Additional information

### Funding

| Funder | Author |
| --- | --- |
| Natural Sciences and Engineering Research Council of Canada | Saeedeh Hashemnia<br>David R Euston<br>Aaron J Gruber |
| Beswick Foundation | Saeedeh Hashemnia |

The funders had no role in study design, data collection and interpretation, or the decision to submit the work for publication.

## Author contributions
Saeedeh Hashemnia, Data curation, Formal analysis, Investigation, Visualization, Writing - original draft; David R Euston, Conceptualization, Methodology, Writing - review and editing; Aaron J Gruber, Conceptualization, Resources, Supervision, Funding acquisition, Writing - original draft, Writing - review and editing

## Author ORCIDs
Aaron J Gruber ⓘD https://orcid.org/0000-0003-2700-5429

## Ethics
Animal experimentation: All procedures were performed in accordance with the Canadian Council of Animal Care and the Animal Welfare Committee at the University of Lethbridge (AWC# 1512).

## Decision letter and Author response
Decision letter https://doi.org/10.7554/eLife.56755.sa1
Author response https://doi.org/10.7554/eLife.56755.sa2

# Additional files

## Supplementary files
• Transparent reporting form

## Data availability
Data and analysis code are available online (https://github.com/SaeedehUleth/AMPH-and-utility-encoding; copy archived at https://github.com/elifesciences-publications/AMPH-and-utility-encoding).

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
