## [Decision Letter]

**Acceptance summary:**

This is an exciting paper in which the authors examine the effects of low and high dose amphetamine on behavior and single unit activity in ACC in rats performing the classic t-maze task used to test for effects of effort on value-based choice. Willingness to extert effort depends on ACC and VS and dopamine function, and has been shown to be modified by amphetamine. Specifically, willingness to exert effort is increased at low doses, despite paradoxical reductions in apparent reward value. In the current manuscript the authors replicate this effect behaviorally, also showing that higher doses impair performance. Then they show that neurons recorded in the ACC alter their encoding of utility – defined as the joint encoding of effort and value – in a way that explains these results. Through novel analyses they show that despite relatively little change in the separate encoding of effort and value, the joint representation of these variables becomes steeper, due to compression of encoding of reward value, then disrupted with amphetamine. Accompanying this, they also show that ensemble movement in the state space defining the task becomes more stereotyped at low doses, consistent with acute effects of amphetamine and with models of tonic dopamine effects on prefrontal ensemble trajectories. Overall, the results provide a compelling window into the mechanism of action of amphetamine on prefrontal function and associated behavior.

**Decision letter after peer review:**

Thank you for submitting your article "Amphetamine reduces utility encoding and stabilizes neural dynamics in rat anterior cingulate cortex" for consideration by *eLife*. Your article has been reviewed by three peer reviewers, including Geoffrey Schoenbaum as the Reviewing Editor and Reviewer #1, and the evaluation has been overseen by Kate Wassum as the Senior Editor. The following individuals involved in review of your submission have agreed to reveal their identity: Stephanie Mary Groman (Reviewer #2).

The reviewers have discussed the reviews with one another and the Reviewing Editor has drafted this decision to help you prepare a revised submission.

We would like to draw your attention to changes in our revision policy that we have made in response to COVID-19 (https://elifesciences.org/articles/57162). Specifically, when editors judge that a submitted work as a whole belongs in *eLife* but that some conclusions require a modest amount of additional new data or analyses, as they do with your paper, we are asking that the manuscript be revised to either limit claims to those supported by data in hand, or to explicitly state that the relevant conclusions require additional supporting data.

Our expectation is that the authors may eventually carry out the additional experiments and report on how they affect the relevant conclusions either in a preprint on bioRxiv or medRxiv, or if appropriate, as a Research Advance in *eLife*, either of which would be linked to the original paper.

Summary:

This is an exciting paper in which the authors examine the effects amphetamine on behavior and single unit activity in ACC in rats performing the classic t-maze task used to test for effects of effort on value-based choice. As expected, willingness to exert effort is increased at low doses, despite paradoxical reductions in apparent reward value. Higher doses impaired performance. Analyses of single unit encoding of effort and reward quantity showed that the joint representation of these variables changes consistent with this behavioral effect, initially due to compression of encoding of reward value and tighter "scripting" of state space trajectories, followed by disorganization at high doses. Overall, the results provide a compelling window into the mechanism of action of amphetamine on prefrontal function and associated behavior.

Essential revisions:

The reviewers were agreed that the revisions could be accomplished with new analyses and changes to the text. The essential changes fall into 4 categories:

1) There were areas of confusion or lack of detail on methods. Fixing these should be straightforward, but must be done. This covers many of the issues raised by reviewer 2 and some by reviewer 3.

2) Reviewer 2 and reviewer 3 asked for additional information and control analyses to justify some of the statistical approaches used. Items 3 and 5 especially in reviewer 3 are key and were echoed by other reviews. Providing additional detail and discussing implications of how these questions fall out is essential.

3) Please clarify how the key analyses of the correlated activity was done. This issue is important for reviewer 3, point 4 and reviewer 1 was also interested in whether the changes in correlated activity reflects a loss of joint encoding (utility) or is an outcome of changes in one category (reward).

4) All reviewers agreed that some discussion of the drug effects and their dependence on dose and chronicity would be warranted. Since the use of an ABA design is not possible, there is a confound between timing and drug dose that should be mentioned. Also, since the design is chronic, some of the effects might be related to repeated administration or sensitization – that is, they might not appear with acute administration initially. At least that seems the case to the reviewers. This should be addressed either on rebuttal or with new analyses or in discussion.

Reviewer #1:

This is an exciting paper in which the authors examine the effects of low and high dose amphetamine on behavior and single unit activity in ACC in rats performing the classic t-maze task used to test for effects of effort on value-based choice. As noted, willingness to extert effort depends on ACC and VS and dopamine function and has been shown to be modified by amphetamine. Specifically, willingness to exert effort is increased at low doses, despite paradoxical reductions in apparent reward value. In the current manuscript the authors replicate this effect behaviorally, also showing that higher doses impair performance. Then they show that neurons recorded in the ACC alter their encoding of utility – defined as the joint encoding of effort and value – in a way that explains these results. Through novel analyses they show that despite relatively little change in the separate encoding of effort and value, the joint representation of these variables becomes steeper, due to compression of encoding of reward value, then disrupted with amphetamine. Accompanying this, they also show that ensemble movement in the state space defining the task becomes more stereotyped at low doses, consistent with acute effects of amphetamine and with models of tonic dopamine effects on prefrontal ensemble trajectories. Overall, the results provide a compelling window into the mechanism of action of amphetamine on prefrontal function and associated behavior.

Overall, I very much liked the paper. I thought the hypothesis was clear, the experiment tested it directly and the results were well described and generally compelling. Indeed, I only have one real suggestion, which is that the authors might add some analyses of joint representation of effort and value to their core analysis in Figure 4. While I very much like this part of the paper, it seems to me that a large fraction of the units plotted in the scatters are likely non-selective. And the prior analysis suggests that there is little effect on activity related to reward or effort in isolation. It seems to me if I understand the core point here that this plot might then suggest that the main effect of amphetamine is to initially improve and then disrupt neurons that jointly represent both effort and reward. If neurons with significant correlations in both domains (or significant main effects of utility in a 2-factor ANOVA) were plotted separately and quantified, the plots might become substantially clearer and there would be a dramatic and selective loss of such neurons. It would be interesting to know if this occurs against a backdrop of preserved isolated coding of reward size and effort.

So that is my main question in reading the results. Overall, I thought it was a terrific paper.

Reviewer #2:

The manuscript titled "Amphetamine reduces utility encoding and stabilizes neural dynamics in rat anterior cingulate cortex" by Hashemnia and colleagues describes an experiment aimed at understanding how amphetamine impacts utility encoding in the anterior cingulate cortex. The authors record neural dynamics in the ACC of rats concurrently performing a forced effort/reward task before and after systemic administration of amphetamine. The authors argue that amphetamine reduced reward signaling but increased stability of ensemble dynamics in the ACC which explains why administration of amphetamine increases effort expenditure but reduces reward intake.

I think the results could be of broad interest, but the Materials and methods section and Results section of the paper are incredibly dense. I think that the analytical approaches described here will not be digestible to most readers. Moreover, inadequate descriptions of key experimental procedures do not enable a proper evaluation of the results and/or interpretation provided by the authors. I have provided some comments below:

1) Figure 1 suggests that all trials were forced, so how can the authors be sure that administration of amphetamine biases rats to the high-reward/high-effort condition that they were trying to model? Data demonstrating that they are able to get the effect they are trying to understand is critical.

2) What does a session consist of? I assume it is 5 blocks of 20 trials for each of the conditions, but were they always performed in the same order (e.g., reward manipulation first followed by effort manipulation). If so, this could be problematic for the analysis as the manipulation covaries with satiation state of the animal.

3) Along these same lines, if the blocks were always conducted in the same order how can the authors be confident that the effect of amphetamine is consistent across the blocks? Because the duration of each session/block is unknown, the different effects of amphetamine on session blocks could just due to pharmacokinetic changes.

4) The authors need to explain why they did not counterbalance the order of amphetamine dose in the majority of their rats and state how many days (if any) separated each dose. When were the saline conditions performed?

Reviewer #3:

This is an interesting and well-conceived study that attempts to determine why stimulant use leads to an increase in task engagement while simultaneously reducing reinforcer drive. Neural recordings were made from the ACC of rats performing an effort-reward task while receiving increasing doses of d-amphetmaine (AMPH). An elegant set of analyses examine how utility encoding at the neuronal level is influenced by increasing doses of AMPH. Additionally, population encoding is assessed via state space analyses in reduced spaces via GPFA. Several inferences are made about the dose-dependent effects of AMPH on neural activity: Low doses increase excitability and stability in a manner optimal for task execution, whereas high doses reduce utility encoding and create excessive time-lagged correlations that may reflect the over stabilization of task-related features in ACC networks.

I have the following concerns:

1) The conclusions regarding dose are confounded with repeated administration and the time scale of administration. Animals rapidly and robustly sensitize to repeated doses of AMPH. The inferences here made about dose may reflect neuroadaptations to the prior dose. In addition, it's difficult to interpret the effects of AMPH as minimal info is provided about the timing of the dose relative to ongoing behavior. How long was neural activity recorded after injection? Monoamine release during this time is a moving target and therefore early trials may not be comparable to late ones. Did animals receive more than one daily session at a given dose (e.g. > 1 injection of 0.5 mg/kg)?

2) Lapish et al., (2015) observed a robust change in firing rate following AMPH administration. To what extent was this observed in the current study and did it influence any of the utility or state space analysis?

3) At the heart of the utility analysis is the Pearsons correlation between two categories of effort or reward. If the predictor is dichotomous (low, high), I am not sure how Pearson is appropriate here. I am concerned that this confounds the subsequent analyses based on this correlation.

4) Further, the specifics of the analysis of the joint encoding is lacking and would benefit from a more formal (e.g. formulaic) presentation. Inference is made by the change in the slope of the first PC about the rotation towards reward. However, how can we be sure that this rotation does not simply reflect the breakdown of utility encoding? From the images in 4A, it does not appear that there is any expansion in the vertical/effort axis, which one might expect if there was truly rotation towards it.

5) A caveat of the current study is that GPFA is a statistical procedure to reduce the dimensions of the recorded neural population but may not provide an accurate or optimal representation of the underlying state spaces. It is possible that reductions in the volume of the spaces might reflect the distribution of variance to other, not assessed, components. Can this be accounted for? Perhaps assessing the decay of the eigenvalues could help here.

6) The term "scripts" is used a couple of times throughout the text. It seems that, conceptually, this is related to a basin of attraction where neural activity can unfold. Further, since the results herein seem to be consistent with Lapish et al., (2015), to what extent can the "scripts" be related to basins of attraction?

---

## [Author Response]

Essential revisions:Reviewer #1:This is an exciting paper in which the authors examine the effects of low and high dose amphetamine on behavior and single unit activity in ACC in rats performing the classic t-maze task used to test for effects of effort on value-based choice. As noted, willingness to extert effort depends on ACC and VS and dopamine function and has been shown to be modified by amphetamine. Specifically, willingness to exert effort is increased at low doses, despite paradoxical reductions in apparent reward value. In the current manuscript the authors replicate this effect behaviorally, also showing that higher doses impair performance. Then they show that neurons recorded in the ACC alter their encoding of utility – defined as the joint encoding of effort and value – in a way that explains these results. Through novel analyses they show that despite relatively little change in the separate encoding of effort and value, the joint representation of these variables becomes steeper, due to compression of encoding of reward value, then disrupted with amphetamine. Accompanying this, they also show that ensemble movement in the state space defining the task becomes more stereotyped at low doses, consistent with acute effects of amphetamine and with models of tonic dopamine effects on prefrontal ensemble trajectories. Overall, the results provide a compelling window into the mechanism of action of amphetamine on prefrontal function and associated behavior.Overall, I very much liked the paper. I thought the hypothesis was clear, the experiment tested it directly and the results were well described and generally compelling. Indeed, I only have one real suggestion, which is that the authors might add some analyses of joint representation of effort and value to their core analysis in Figure 4. While I very much like this part of the paper, it seems to me that a large fraction of the units plotted in the scatters are likely non-selective. And the prior analysis suggests that there is little effect on activity related to reward or effort in isolation. It seems to me if I understand the core point here that this plot might then suggest that the main effect of amphetamine is to initially improve and then disrupt neurons that jointly represent both effort and reward. If neurons with significant correlations in both domains (or significant main effects of utility in a 2-factor ANOVA) were plotted separately and quantified, the plots might become substantially clearer and there would be a dramatic and selective loss of such neurons. It would be interesting to know if this occurs against a backdrop of preserved isolated coding of reward size and effort.So that is my main question in reading the results. Overall, I thought it was a terrific paper.

Thank you for the excellent suggestion, which prompted us to dig deeper into the data. Analyzing utility cells independent from other types is complicated by the fact that some neurons lose their reward sensitivity whereas others gain it after AMPH. We therefore retained (and extended) the analysis of the distribution of all cells in the effort-reward plane without selection criteria so as to capture all of these effects. So, to implement a version of your suggestion we now identify the proportion of neurons that significantly signal reward, and independently compute the proportion signaling effort. This captures the many cells that are not jointly encoding both effort and reward. We find that the proportion of reward cells strongly decreases with increasing AMPH, whereas the number of effort cells does not (Figure 4D-E). This is further evidence that AMPH selectively impairs reward signaling. It also shows that the change is not just restricted to utility cells, but is a general attenuation of reward signaling, as described in the next paragraph.

The next question we now address is why fewer neurons discriminate reward volume under AMPH. It could be that the cells lose their signal, which is the difference of firing rate between conditions of high and low reward. Or it could be that the signal amplitude is the same, but that increased variation (e.g. noise) obscures discrimination. We now show how these change following AMPH, which reveals an interesting dose-dependent phenomenon. The signal significantly reduces with AMPH (Figure 4F). The variation, however, shows a different profile (Figure 4G). Although not significantly different by ANOVA, the variation around the mean is reduced at 1.0mg/kg, but then increases at 1.5mg/kg. This is consistent with our newly added analysis of the variation in ensemble neural trajectories, which is lower than saline at intermediate doses, but higher at 1.5mg/kg. These additions support our new focus in the manuscript on network stability.

Reviewer #2:The manuscript titled "Amphetamine reduces utility encoding and stabilizes neural dynamics in rat anterior cingulate cortex" by Hashemnia and colleagues describes an experiment aimed at understanding how amphetamine impacts utility encoding in the anterior cingulate cortex. The authors record neural dynamics in the ACC of rats concurrently performing a forced effort/reward task before and after systemic administration of amphetamine. The authors argue that amphetamine reduced reward signaling but increased stability of ensemble dynamics in the ACC which explains why administration of amphetamine increases effort expenditure but reduces reward intake.I think the results could be of broad interest, but the Materials and methods section and Results section of the paper are incredibly dense. I think that the analytical approaches described here will not be digestible to most readers. Moreover, inadequate descriptions of key experimental procedures do not enable a proper evaluation of the results and/or interpretation provided by the authors. I have provided some comments below:

We have extensively revised the Methods & Results to better describe methodology in a way that is more accessible to a wide audience. We identify changes to address specific deficits revealed by the reviewers’ comments below.

1) Figure 1 suggests that all trials were forced, so how can the authors be sure that administration of amphetamine biases rats to the high-reward/high-effort condition that they were trying to model? Data demonstrating that they are able to get the effect they are trying to understand is critical.

We now see how our previous introduction was mismatched with our experiment. We did not intend to model the change in choice bias. Our aim in this study is to better understand the effects of AMPH on the signaling of reward, effort, and task engagement in the ACC, as a precursor to understanding how AMPH affects the neural basis of free choice. We now specify more clearly in the Abstract & Intro that we aim to better understand the apparently paradoxical effect in which AMPH evokes greater task engagement and effort, while also reducing reward consumption (rather than the mechanisms of choice among options).

We used a forced alternation design to reduce errors from unequal sampling and confounds. Well-performing rats will not freely generate many responses to low utility options, so we will be under powered in statistically testing for differences of neural signaling in this condition.

Although rats do not engage in choice among options in the present task, the behavioural evidence shows that AMPH is exerting several of its hallmark effects. Animals run faster but spend less time on reward consumption. Therefore, we believe that our report is relevant to the large body of literature on psychostimulant effects on choice and vigilance.

We have extensively edited the Abstract, Introduction and other sections to specify that our aim was not to investigate the effects of AMPH on the neural mechanisms of choice per se, but rather encoding of effort and reward that are important in this choice. Specifically:

Abstract: Rewritten to focus on the dissociated effects on AMPH on task engagement and reward consumption.

Introduction: Edited to focus on dissociated effects as in the Abstract. We now state: “Here, we attempt to link these independent observations to better understand how AMPH affects ACC encoding and dynamics pertinent to task engagement and outcome valuation.”

Results section: We have changed the presentation of the data in Figure 5, Figure 6 and Figure 7 to illustrate how the ensemble dynamics relate to task features and task engagement, and then how these change after AMPH.

Discussion section: We note that the choice here is between engaging in the task, or not performing the task: “It therefore seems that animals track upcoming effort and reward, even though they are not able to choose among the target feeders. Their choice in the present experiment is whether to engage in the task or not. Our data are therefore more revealing of mechanisms of task engagement than choice. Indeed, the present data do not provide conclusive evidence as to why animals shift their preference to high-effort high-reward options under low dose AMPH (Floresco et al., 2008b; Bardgett et al., 2009). Effort encoding was not affected by AMPH, so it does not appear that the brain discounts effort or cannot discriminate future effort levels.”

2) What does a session consist of? I assume it is 5 blocks of 20 trials for each of the conditions, but were they always performed in the same order (e.g., reward manipulation first followed by effort manipulation). If so, this could be problematic for the analysis as the manipulation covaries with satiation state of the animal.

We have revised the manuscript in several places to describe the task design and the rationale (Materials and methods section; Results section) and updated the illustration of the task design (Figure 1B) in attempt to increase clarity. The reviewer is correct that the blocks are in the same order in every session and may present a satiation confound. This was intentional, and due to pragmatic constraints. First, our experience with effort-reward tasks in Fisher-Brown Norway rats is that some implanted animals will not perform trials if faced with barrier climbs early in the session. Second, the resources required to fully counterbalance the task parameters and treatments (block order, drug dose sequence, drug delivery order) are prohibitively expensive. We therefore used other experimental design features to control for confounds. First, we use a within-session contrast to compensate for changes between sessions. We do observe some minor effects after injection of vehicle, which are likely due to changes in motivation. In order to control for motivational changes within sessions, we used several doses of drug (and vehicle). We observed dose-response effects, and these were often non-linear. They showed a ‘U’ or inverted ‘U’ shape. This is unlikely to come primarily from the lower motivation. So, although motivation is likely lower in the second half of the session in which we give drug, the motivation is likely similar from session to session such that it does not obscure dose-dependent changes.

We now describe the confounds (block order, and repeated administrations as mentioned below), and the experimental designs employed to account for them, in the Materials and methods section and Discussion section: “Furthermore, we used a before/after design in which the drug was always given in the second half of the experiment. This allowed us to observe the effect of AMPH on signaling of an identified set of cells. However, any consistent changes that occur during the session, such as satiation, may therefore confound the results. The between-session contrasts help mitigate this confound. The pattern of results we observed, particularly the fact that responses often had different polarity for high vs low doses, suggests that this potential confound is not strongly influencing our results or interpretations.”

3) Along these same lines, if the blocks were always conducted in the same order how can the authors be confident that the effect of amphetamine is consistent across the blocks? Because the duration of each session/block is unknown, the different effects of amphetamine on session blocks could just due to pharmacokinetic changes.

That is possible. However, the second half of the trials were typically completed within 25 minutes (see Author response image 1). This began 10 minutes after AMPH injection, so nearly all trials were completed within 35 minutes of injection. The half life of d-amphetamine in the brain following i.p. injection is 1.2 +/- 0.1 hours (Lokiec et al., 1978). Therefore, our data are collected with ~50% of the half-life, so the concentration should be relatively high throughout testing. Nonetheless, the concentration is expected to be lower at the end of the session. Because we used several doses of AMPH, we would still expect to see a dose-response effect if AMPH is affecting effort. The fact that we found no appreciable covariation of AMPH and effort encoding suggests that this possible confound of the design is not relevant to our interpretation of the data.

4) The authors need to explain why they did not counterbalance the order of amphetamine dose in the majority of their rats and state how many days (if any) separated each dose. When were the saline conditions performed?

Each dose (or saline) occurred on different days. Some saline sessions are intermixed between the AMPH sessions in attempt to capture some portion of any lasting adaptations due to AMPH administration.

We used escalating dosages in attempt to minimize sensitization and any lasting neural changes that are dose-dependent. This rationale is now presented in Materials and methods section and Discussion section. Although counterbalancing is the best design in many scenarios, it is not obvious that it is optimal here. In our experience, many drug naïve rats will not perform some tasks if the first exposure to a psychoactive drug is a high dose. For the present task, we are running near the ceiling at 1.5mg/kg in Fisher-Brown Norway rats that had previously experienced lower doses. At higher doses (2.0 mg/kg), even these AMPH-experienced animals did not perform enough trials for analysis. Because of the sensitization mentioned by reviewers, we suspect that some animals may not even work at 1.5 mg/kg if this dose was their first exposure to AMPH. Moreover, we don’t know what lasting effect higher doses of AMPH may have, or if the effect depends on whether or not the rat performs the task while on the drug. We could address this by counterbalancing over a large number of animals, but this is impractical because of the considerable resources needed for present techniques to record ensemble activity in behaving rats.

The number of days between recording sessions varies because some sessions had low unit yield. Each experimental day, we evaluate the number and quality of units available. If the yield is acceptable, we perform the recording experiment. We then adjust the depths of electrodes and return the animal to the home cage, where it remains until the next session (1-2 days). This procedure is standard in the field, and increases the stability of unit waveforms during the task, as compared to turning electrodes just prior to the task. We have edited the Materials and methods section to clarify the drug schedule and specify that only one drug dose is given per day.

Reviewer #3:This is an interesting and well-conceived study that attempts to determine why stimulant use leads to an increase in task engagement while simultaneously reducing reinforcer drive. Neural recordings were made from the ACC of rats performing an effort-reward task while receiving increasing doses of d-amphetmaine (AMPH). An elegant set of analyses examine how utility encoding at the neuronal level is influenced by increasing doses of AMPH. Additionally, population encoding is assessed via state space analyses in reduced spaces via GPFA. Several inferences are made about the dose-dependent effects of AMPH on neural activity: Low doses increase excitability and stability in a manner optimal for task execution, whereas high doses reduce utility encoding and create excessive time-lagged correlations that may reflect the over stabilization of task-related features in ACC networks.I have the following concerns:1) The conclusions regarding dose are confounded with repeated administration and the time scale of administration. Animals rapidly and robustly sensitize to repeated doses of AMPH. The inferences here made about dose may reflect neuroadaptations to the prior dose. In addition, it's difficult to interpret the effects of AMPH as minimal info is provided about the timing of the dose relative to ongoing behavior. How long was neural activity recorded after injection? Monoamine release during this time is a moving target and therefore early trials may not be comparable to late ones. Did animals receive more than one daily session at a given dose (e.g. > 1 injection of 0.5 mg/kg)?

We have extensively revised the Materials and methods section in attempt to improve clarity. As described above, animals received only one dose per day, and we have addressed the within-session pharmacokinetics in our response to C5.

Sensitization and neuradaptations are certainly a concern. We acknowledge sensitization and its reported effects on effortful behavior, and present a consolidated description of how our task design and analysis attempts to mitigate possible confounds (subsection “Experiment”): “Rats quickly sensitize to AMPH, and withdrawal from repeated administration of high doses can affect effort-reward behavior (Hart et al., 2018). We thus employed four experiment design elements in attempt to control for the potential confound of sensitization and/or withdrawal: (1) we used a within-session design so that the pre-drug blocks serve as a control for lasting changes due to prior drug administration; (2) sessions were separated by at least 24 hours to provide some recovery time; (3) saline control sessions were collected on days between AMPH sessions so as to include some of any lasting changes from repeated AMPH; and (4) we used an escalating drug schedule, rather than counterbalancing dose order, for most animals. Three doses of AMPH (0.5, 1, 1.5 mg/kg) were used in this study. Only one dose (or vehicle) was given per daily session. The order of drug/saline injection across sessions was: vehicle, AMPH 0.5 mg/kg, AMPH 1 mg/kg, vehicle, AMPH 1.5 mg/kg. One rat received vehicle, AMPH 0.5 mg/kg, AMPH 1.5 mg/kg, vehicle, AMPH 1 mg/kg, vehicle, AMPH 0.5 mg/kg. The rationale for this schedule is twofold. First, we have observed that rats often will not perform operant tasks if their first exposure to AMPH is a high dose. Secondly, escalating doses minimize the cumulative amount of prior AMPH exposure on each testing day, which likely minimizes the amount of neural adaptation.”

We also revisit this in the final paragraph of the Discussion section: ”One caveat to our study is that we used repeated injections of AMPH to provide escalating doses in the same set of animals. We used this design because….”

2) Lapish et al., (2015) observed a robust change in firing rate following AMPH administration. To what extent was this observed in the current study and did it influence any of the utility or state space analysis?

We have added a plot of the change in mean firing rate (Figure 3C). AMPH does increase firing rate, relative to lower doses and the pre-drug blocks. The state-space analysis should be relatively insensitive to baseline shifts of mean rate. This is because the correlation metrics for single unit analysis depend on the variance of the signal during the task, but not the absolute magnitude of the signal. We computed the volume of state space occupancy in such a way that it is invariant to translations of the hull within the space. Volume should, therefore, be relatively insensitive to baseline shifts of firing rate, while remaining sensitive to variation of signaling. Indeed, the dose-response profiles for firing rate and state-space volume are different (monotonic vs. parabolic), suggesting that changes in firing rates (which are monotonically increasing) do not fully account for changes in ensemble dynamics.

3) At the heart of the utility analysis is the Pearsons correlation between two categories of effort or reward. If the predictor is dichotomous (low, high), I am not sure how Pearson is appropriate here. I am concerned that this confounds the subsequent analyses based on this correlation.

We apologize for not explaining the rationale for this analysis in the manuscript. The Person correlation is embedded in linear regression analysis, which is one of the most common methods used to quantify the relationship between a binary (categorical) variable and a continuous one. Our analysis is equivalent to a linear regression analysis of firing rate with one binary predictor (high/low reward, or high/low effort). We now explicitly state this in the Materials and methods section: “We use an approach following linear regression analysis, which is the standard method for detecting the relationship between binary predictor variables (reward and effort in the present case) and a continuous variable (firing rate)”. We make a similar statement in the Results section.

Furthermore, we now present evidence using ANOVA on firing rates to detect neurons discriminating effort or reward. This analysis corroborates that of the correlation analysis, suggesting that it is not an artifact of the methodology.

4) Further, the specifics of the analysis of the joint encoding is lacking and would benefit from a more formal (e.g. formulaic) presentation. Inference is made by the change in the slope of the first PC about the rotation towards reward. However, how can we be sure that this rotation does not simply reflect the breakdown of utility encoding? From the images in 4A, it does not appear that there is any expansion in the vertical/effort axis, which one might expect if there was truly rotation towards it.

This is an excellent point. We show (now Figure 4C) the amount of variance explained by the first PC, which is a direct measure of the dispersion of the point spread around this axis. A breakdown in utility would therefore decrease the explained variance by the first PC, which is the case for 1.0 and 1.5 mg/kg. This does not tell us whether it is the effort or reward component that is responsible. We explicitly state this in the Results section. As described above in response to reviewer 1, we now include an analysis of reward encoding and effort encoding cells to provide more insight into which variable is more affected, which shows that it is a reduction in reward signaling that is most likely responsible.

5) A caveat of the current study is that GPFA is a statistical procedure to reduce the dimensions of the recorded neural population but may not provide an accurate or optimal representation of the underlying state spaces. It is possible that reductions in the volume of the spaces might reflect the distribution of variance to other, not assessed, components. Can this be accounted for? Perhaps assessing the decay of the eigenvalues could help here.

This is a good point. We now state in the Results section: “Similar to PCA, GPFA serves to capture as much variance as possible and does not optimize for any particular information present in the data (e.g. reward, location).”

Because the variance important for discriminating task features might be in factors of lower rank that the first 3 we analyze, we have removed our previous quantification of reward and effort encoding in the reduced space.

In order to evaluate if variance has shifted to lower ranking factors after AMPH, we now quantify the change in mean explained variance by the first 3 factors (Figure 7D; subsection “AMPH contracts ACC state-space occupancy at low doses, and expands it at high doses”). The differences do not deviate from 0 for any dose. We chose this format over analysis of eigenvalue decay because it seemed easier to convey to a broad audience

6) The term "scripts" is used a couple of times throughout the text. It seems that, conceptually, this is related to a basin of attraction where neural activity can unfold. Further, since the results herein seem to be consistent with Lapish et al., (2015), to what extent can the "scripts" be related to basins of attraction?

Yes, these concepts are very much related. We apologize that the original draft of the manuscript did an exceedingly poor job of linking the data to previous work and proposals of attractors. This is even more important now that we have focused the manuscript on analysis of variance at the single unit and ensemble levels that inform on stability. We now devote two sizeable paragraphs in the Discussion section to explain how our results cohere with that of Lapish and prior computational work on the neuromodulation of attractors in the mPFC. Briefly, we have evidence of dose-dependent stability, where low doses increase stability and high doses decrease it. This is consistent with Lapish. Where we differ, is that Lapish and previous papers from that group find evidence of clusters in state space for distinct epochs (indicative of point attractors), whereas we find a smooth trajectory linking them (indicative of attractors with higher dimensionality, like lines or surfaces). This could be due to the number of units recorded, task demands, or dimensionality reduction procedure. We point out that the GPFA does not optimize for any specific encoding. In any case, we argue that the smooth trajectory reflects an unfolding of a mental ‘script’ of task execution that not only encodes the sequence, but also the proximity in time and space. This is inherently linked to the notion of attractors, and compatible with the interpretation of Lapish. Our proposal goes a little further and suggests that at least some components of ACC activity has resolution beyond epochs defined by periods between task events.